# Straightforward preparation of supramolecular Janus nanorods by hydrogen bonding of end-functionalized polymers

Shuaiyuan Han [1], Sandrine Pensec [1], Dijwar Yilmaz [1], Cédric Lorthioir [2], Jacques Jestin[3], Jean-Michel Guigner[4], Frédérick Niepceron[5], Jutta Rieger [1], François Stoffelbach[1], Erwan Nicol[5], Olivier Colombani [5✉] & Laurent Bouteiller [1✉]

Janus cylinders are one-dimensional colloids that have two faces with different compositions and functionalities, and are useful as building blocks for advanced functional materials. Such anisotropic objects are difficult to prepare with nanometric dimensions. Here we describe a robust and versatile strategy to form micrometer long Janus nanorods with diameters in the 10-nanometer range, by self-assembly in water of end-functionalized polymers. The Janus topology is not a result of the phase segregation of incompatible polymer arms, but is driven by the interactions between unsymmetrical and complementary hydrogen bonded stickers. Therefore, even compatible polymers can be used to form these Janus objects. In fact, any polymers should qualify, as long as they do not prevent co-assembly of the stickers. To illustrate their applicative potential, we show that these Janus nanorods can efficiently stabilize oil-in-water emulsions.

[1] Sorbonne Université, CNRS, Institut Parisien de Chimie Moléculaire, UMR 8232, Equipe Chimie des Polymères, 75252 Paris, France. [2] Sorbonne Université, CNRS, Laboratoire de Chimie de la Matière Condensée de Paris, UMR 7574, 75252 Paris, France. [3] Laboratoire Léon Brillouin, UMR12 CEA-CNRS, Bât. 563, CEA Saclay, 91191 Gif-sur-Yvette, France. [4] Sorbonne Université, CNRS, Institut de Minéralogie, de Physique des Matériaux et de Cosmochimie, UMR 7590-IRD-MNHN, 75252 Paris, France. [5] Institut des Molécules et Matériaux du Mans (IMMM), UMR 6283 CNRS Le Mans Université, Avenue Olivier Messiaen, 72085 Le Mans Cedex 9, France. ✉email: olivier.colombani@univ-lemans.fr; laurent.bouteiller@sorbonne-universite.fr

Janus particles[1,2] are non-centrosymmetric objects that have two faces with different compositions and functionalities. The resulting asymmetry provides an opportunity to encode them with properties[3–6] such as controlled motion or actuation, or to direct their self-assembly into complex superstructures[7–12]. A wide variety of preparation methods exist to build such particles with micrometer dimensions, but it is still a great synthetic challenge to prepare nanometric Janus particles, in particular with nonspherical shapes[13]. Moreover, the only known approaches[14,15] to form polymeric Janus nanorods are limited to highly incompatible polymers.

Here we describe a robust and versatile strategy to form (micrometer long) Janus nanorods with diameters in the 10-nanometer range, by self-assembly in water of two end-functionalized polymers. The Janus organization results from the unsymmetrical and complementary hydrogen bonded stickers present on both polymers. The obtained Janus topology is therefore independent of the actual polymers used, as long as they do not prevent co-assembly. The versatility and scalability of this approach make possible the investigation of the rich properties that can be predicted for such easily functionalizable nano-objects. In particular, we show that these Janus nanorods are excellent stabilizers for oil-in-water emulsions.

## Results

**Design and synthesis**. Hydrophilic polymers decorated with a bisurea sticker are known to form long rod-like assemblies in water[16–18]. Interestingly, Sijbesma et al. showed that the length of the alkylene spacer connecting two urea groups plays an essential role: if two bisurea stickers differing by the size of the alkylene spacer are mixed, they form self-sorted assemblies, so that urea–urea hydrogen bonds are maximized[19]. By analogy, we designed an unsymmetrical trisurea sticker where the three urea groups are connected by two spacers of unequal lengths (Fig. 1a). We anticipated that this unsymmetrical sticker would be able to self-assemble into long objects, but with a fixed orientation within the assembly, so as to maximize urea–urea hydrogen bonds. Functionalization of this sticker either by a red polymer arm on one side or by a blue polymer arm on the other side yields two components that may co-assemble. In such a co-assembly, the red and blue arms should segregate to avoid mismatched hydrogen bonding of the urea groups, imposing the formation of Janus nanorods, no matter the incompatibility between the polymer arms.

To test this concept, two chain transfer agents have been synthesized based on a trithiocarbonate moiety connected to one or the other side of an unsymmetrical trisurea sticker (see Supplementary Methods). Polymerization of N,N-dimethylacrylamide (DMAc) or N-acryloylmorpholine (NAM) under reversible addition-fragmentation transfer (RAFT) conditions afforded 4/8-PDMAc and 8/4-PNAM (Fig. 1b), where "4/8" and "8/4" refer to the number of $CH_2$ units separating the ureas, and thus to the relative orientation of the unsymmetrical stickers. Number-average degrees of polymerization ($DP_n$) of 32 and 46 were obtained, respectively, with narrow dispersities ($Đ \sim 1.1$).

**Characterization of the co-assembly**. Aqueous dispersions were prepared through a solvent exchange step[20] to favor co-assembly (see Supplementary Fig. 32): the polymers were dissolved in dimethylsulfoxide (DMSO), mixed together and then water was added slowly until a water/DMSO proportion of 99/1. Figure 1c shows that the mixture of 4/8-PDMAc and 8/4-PNAM yields rod-like objects that are several hundreds of nanometers long and have an average diameter of ca. 12 nm. The diffusion coefficients of PDMAc and PNAM chains in the mixture were shown by DOSY NMR (Supplementary Fig. 26) to be the same, which suggests their co-assembly. Actually, co-assembly of (4/8-PDMAc + 8/4-PNAM) and the formation of rod-like objects were confirmed by static light scattering experiments (see Supplementary Fig. 22). 4/8-PDMAc and 8/4-PNAM favor co-assembly instead of self-sorting, probably because of mixing entropy and because of relieved steric hindrance (Fig. 1a). Indeed, the co-assembly provides a distance between two consecutive red (or blue) arms about twice as large as in the self-sorted assemblies (Arguably, this steric bias may not be strong enough to enforce a strict alternation of the blue and red components within the co-assembly, but strict alternation is actually not required to obtain a Janus structure). Interestingly, these objects proved to be stable over months and at least up to 80 °C (see Supplementary Figs. 34 and 35). They also proved to be sufficiently stable to allow dialysis in water and freeze-drying. Cryo-TEM experiment proved that isolating the particles and redispersing them had no obvious effect on their shape (see Supplementary Fig. 29).

**Demonstration of the Janus structure**. Because of the similar contrast of the two polymers in TEM, it is not possible to deduce from the cryo-TEM analyses whether the co-assembled nanorods have a Janus structure or not. Therefore, we designed polymers that should present a suitable contrast in a small angle neutron scattering (SANS) experiment. The partially deuterated PDMAc($D_6$) (Fig. 2a) has a much higher scattering length density ($4.86 \cdot 10^{-6}$ Å$^{-2}$) than the fully hydrogenated PDMAc ($0.94 \cdot 10^{-6}$ Å$^{-2}$). This means that in deuterated water ($6.37 \cdot 10^{-6}$ Å$^{-2}$) the former has a contrast that is 14 times lower than the latter. Accordingly, the co-assembly of 4/8-PDMAc($D_6$) and 8/4-PDMAc was prepared in $D_2O$, and as a control experiment, the same objects but with a homogeneous contrast, were prepared from 4/8-PDMAc and 8/4-PDMAc. As expected, Fig. 2b shows that the mixture of the two fully hydrogenated polymers (4/8-PDMAc + 8/4-PDMAc) yields rod-like objects ($q^{-1}$ dependence at low q). A fit with the form factor of an infinitely long cylinder of elliptical cross-section and homogeneous contrast yields the diameters $d_{minor} = 7.6$ nm and $d_{major} = 12$ nm (The discrepancy between the model and the experiment at high q is due to the fact that the solvated polymer chains at the surface of the objects are not explicitly taken into account by the form factor of a simple cylinder). These values are consistent with the previous microscopy results. Because the molar masses of 4/8-PDMAc($D_6$) and 4/8-PDMAc are the same and because the assemblies were prepared in identical conditions, the assemblies formed from (4/8-PDMAc($D_6$) + 8/4-PDMAc) are expected to have the same overall shape and dimensions as those formed from (4/8-PDMAc + 8/4-PDMAc). The question is how the deuterated chains are distributed within the nanorods. If the deuterated chains were randomly distributed, the cylinders should have the same cross-section as (4/8-PDMAc + 8/4-PDMAc), but with a lower (homogeneous) contrast. Therefore, the normalized scattering data for both experiments should be identical. Supplementary Figure 24 shows that this is not the case. Conversely, if the deuterated chains are segregated in one half of the cylinder, this part of the object becomes virtually invisible to SANS but the other cylinder half should have the same contrast as (4/8-PDMAc + 8/4-PDMAc). Actually, the (4/8-PDMAc($D_6$) + 8/4-PDMAc) data can be fitted quite well with the form factor of an infinitely long cylinder with the same homogeneous contrast and the same minor diameter ($d_{minor} = 7.6$ nm) as (4/8-PDMAc + 8/4-PDMAc), but a much smaller major diameter ($d_{major} = 9.4$ nm) (Fig. 2b). These SANS results definitely prove the nonhomogeneous distribution of the deuterated and hydrogenated chains and are in strong support of the Janus structure.

In addition, NMR spectroscopy experiments were used to probe contacts between polymer chains and thus the Janus

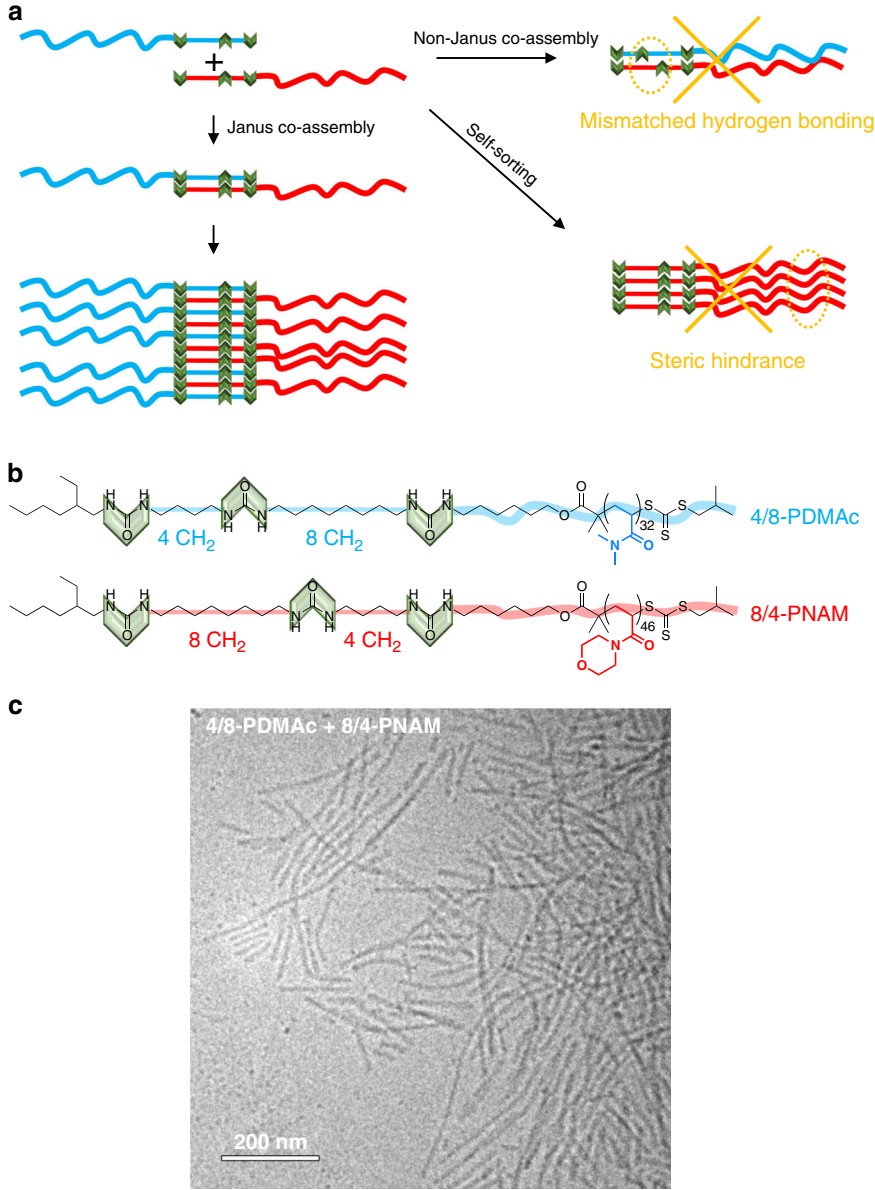

**Fig. 1 Co-assembly. a** Concept: the unsymmetrical stickers constrain the relative orientation of the polymer arms. Co-assembly is favored over self-sorting for entropic and steric reasons, while the Janus organization is favored because it allows optimal matching of the hydrogen bonding units. **b** Structure of 4/8-PDMAc and 8/4-PNAM. **c** Cryo-TEM image of (4/8-PDMAc + 8/4-PNAM) equimolar mixture (1 g L$^{-1}$ in water/DMSO (99/1 v/v)).

topology. First, a 2D $^1$H NOESY experiment was performed. This technique allows probing the spatial proximity between distinct chemical groups in a range typically lower than 4–5 Å. In the present case, Fig. 2c actually shows the absence of any cross-relaxation peaks between PDMAc and PNAM chains in (4/8-PDMAc + 8/4-PNAM) nanorods, which is in agreement with the Janus topology[15]. Secondly, $^1$H transverse relaxation times were measured because they are highly sensitive to segmental motions which are mostly influenced by the local friction and therefore by the local environment of the chains. Figure 2d shows that the $^1$H $T_2$ relaxation signals for the PDMAc methyl groups and the PNAM methylene groups in the (4/8-PDMAc + 8/4-PNAM) co-assembly are identical to those in the respective 4/8-PDMAc and 8/4-PNAM homo-assemblies. This rules out any significant mixing of the two kinds of chains and therefore also supports the Janus topology[21].

In order to complement these scattering and spectroscopic proofs with electron microscopy data, we adapted the system to introduce a polymer with a high affinity for metal ions that may provide contrast in TEM to distinguish between the two sides of the Janus nanorods. Polymerization of an acrylate monomer bearing a crown ether moiety afforded 4/8-PCEA (Fig. 2a). Co-assembly between 4/8-PCEA and 8/4-PDMAc was performed by the usual procedure and one equivalent of silver nitrate (relative to the crown ether) was added to the solution. This solution was deposited on a TEM grid and a negative staining agent (uranyl acetate) was applied. As expected, Fig. 2e shows the presence of rods with a diameter of 9 nm. Close inspection reveals that many of these rods are darker on one side than on the other (Fig. 2e). This is confirmed by a systematic gray scale analysis (see Fig. 2e and Supplementary Fig. 30). In the same conditions, the symmetrical system (4/8-PDMAc + 8/4-PDMAc) shows a perfectly symmetrical contrast (see Supplementary Fig. 31). Therefore, this microscopy experiment affords direct visual evidence of the formation of nanorods with a Janus topology (The fact that some rods in Fig. 2e have a perfectly symmetrical contrast does

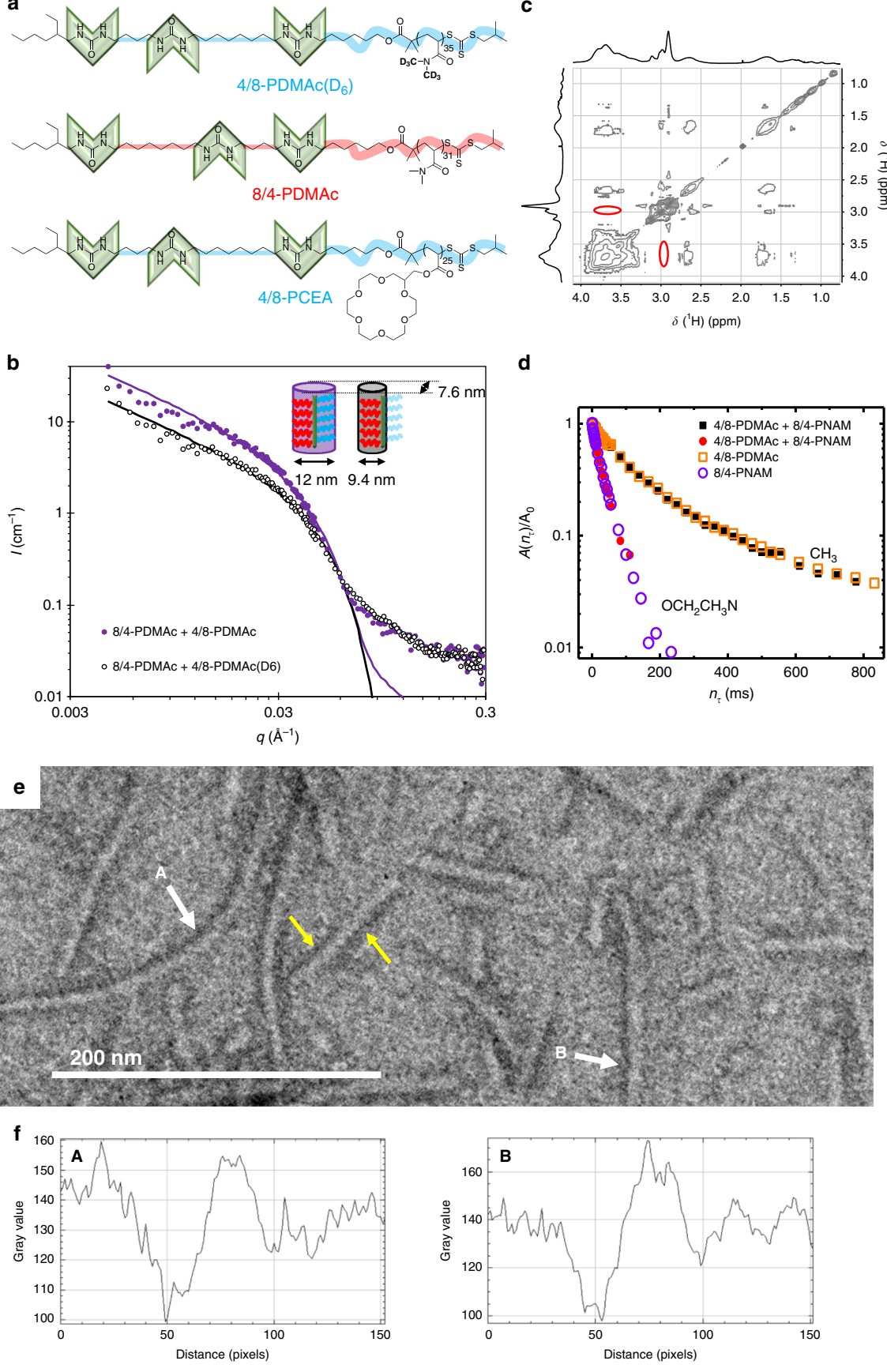

**Fig. 2 Local segregation. a** Structure of 4/8-PDMAc(D$_6$), 8/4-PDMAc and 4/8-PCEA. **b** SANS data of equimolar co-assemblies in D$_2$O (freeze-dried particles, 10 g L$^{-1}$). The curves are fits with the form factor of cylinders of elliptical cross-section that have the same homogeneous contrast. Dimensions of the cross-sections are indicated on the inset (The discrepancy between the model and the experiment at high $q$ is due to the fact that the solvated polymer chains at the surface of the objects are not explicitly taken into account by the form factor of a simple cylinder). **c** $^1$H NOESY NMR contour plot of (4/8-PDMAc + 8/4-PNAM) equimolar mixture in D$_2$O (freeze-dried particles, 5 g L$^{-1}$). The red ellipses highlight the absence of cross-relaxation peaks between the PDMAc and PNAM chains (compare to Supplementary Fig. 27). **d** Normalized $^1$H transverse relaxation signal $A(t)/A_0$ of the PDMAc CH$_3$ protons (squares) and the PNAM OCH$_2$CH$_2$N protons (circles) in the (4/8-PDMAc + 8/4-PNAM) equimolar mixture (solid symbols) or in the pristine polymer (hollow symbols) (1 g L$^{-1}$ in D$_2$O, obtained by dialysis). **e** TEM of (4/8-PCEA + 8/4-PDMAc) equimolar mixture (negative staining). White arrows highlight the PCEA-side of some Janus nanorods. Yellow arrows highlight contrast inversion along the nanorods axis. **f** Gray scale analysis of marked rods in (**e**). See Supplementary Figs. 30 and 31 for more data and control experiment.

**a**

| C : | PDMAc-*b*-PNAM | *block copolymer* |
| D : | 4/8-PDMAc + 8/4-PNAM | *Janus nanorods* |
| E : | 4/8-PDMAc + 8/4-PNAM | *dried & redispersed Janus rods* |
| F : | 4/8-PDMAc + 8/4-PDMAc | *non-Janus nanorods* |

**b**

| N : | P*n*BA-*b*-PDMAc | *block copolymer* |
| O : | 4/8-P*n*BA + 8/4-PDMAc | *Janus nanorods* |

**Fig. 3 Emulsions. a** Water/ethyl acetate (80/20) mixtures with various additives (1 g L$^{-1}$), 2 weeks after shaking. The cloudy phase corresponds to ethyl acetate droplets that have creamed but have not coalesced (see Supplementary Fig. 38). **b** Water/β-pinene (80/20) mixtures with additives (25 mg L$^{-1}$), 2 weeks after vortexing. The letters indicate the nature of the additive (see text).

not mean that non-Janus objects are present, but simply that they are adsorbed with their symmetry plane roughly perpendicular to the TEM grid). These data also reveal a few contrast inversions along the rods axis which suggests some long-range twisting.

**Emulsion stabilization.** With this easily scalable and versatile process in hand we decided to test the properties of the Janus nanorods. Both a Janus topology and an anisotropic shape have been identified as desirable features in the context of interfacial stabilization[22–24]. We therefore tested the stabilization of an oil-in-water emulsion by the 4/8-PDMAc + 8/4-PNAM Janus nanorods. The Janus nanorods were first prepared in water/DMSO (99/1) as described above. Then DMSO was removed by dialysis and the concentration was adjusted as desired. Ethyl acetate (Ethyl acetate was selected as organic phase because it dissolves 4/8-PDMAc but not 8/4-PNAM) was added, and after shaking, the biphasic mixtures were monitored over time. Figure 3a and Supplementary Fig. 37 show that after 2 weeks of storage, ethyl acetate droplets are still present for concentrations of Janus nanorods above 0.5 g L$^{-1}$ (sample D). Confocal microscopy shows that stable ethyl acetate droplets are formed in the range 10–100 µm and that their coalescence is strongly hampered over at least 9 days (Supplementary Fig. 38). In the same conditions, the corresponding PDMAc-*b*-PNAM diblock copolymer (sample C) is not efficient: coalescence of the droplets is nearly quantitative after 2 days. To check if that encouraging result is due to the anisotropy of the objects or to the Janus topology, a blank experiment was performed with homogeneous nanorods prepared by the same process but from a (4/8-PDMAc + 8/4-

PDMAc) mixture (i.e., the same polymer on both sides). Supplementary Figure 38 shows that, unlike the Janus nanorods, the homogeneous nanorods of PDMAc (sample F) do not afford the initial formation of small and rather monodisperse droplets. Therefore, both the cylindrical shape of the particles and their Janus topology seem to be of importance. We then tested the Janus nanorods that had previously been isolated as a dry powder (see above) and introduced them directly in a water/ethyl acetate mixture (sample E). Satisfyingly, the results are very similar to the Janus nanorods that were prepared in situ (i.e., not isolated, sample D). This has an obvious practical interest.

Of course, improved interfacial stabilization is expected if the nature of the polymers is adapted to the polarity of the phases. We therefore, extended our approach to amphiphilic Janus nanocylinders consisting of poly(*n*-butyl acrylate) (P*n*BA) and PDMAc arms by co-assembling 8/4-PDMAc and 4/8-P*n*BA (see Supplementary Methods and Notes for details). Such Janus nanoparticles (Fig. 3b, sample O) were shown to stabilize β-pinene (β-pinene is a bio-based, hydrophobic liquid of industrial relevance in the perfume industry)/water emulsions better than the corresponding PDMAc-b-P*n*BA diblock copolymer (sample N).

## Discussion

We now have a very simple procedure to prepare a wide variety of Janus nanorods in water or in organic solvents. The existing approaches to prepare similar Janus nanorods entail either the synthesis of heterografted cyclic peptides[15] or of triblock copolymers[14], whereas our procedure only requires the synthesis of

two end-functionalized homopolymers. Moreover, the existing approaches are only successful if the two polymer arms are strongly incompatible. In our procedure, microphase separation within the nanorods is the consequence of the stickers co-assembly. It does not have to be the driving force, as shown by the segregation of the hydrogenated and deuterated arms. Finally, because of the versatility of the RAFT polymerization, changing the nature of the polymer arms is straightforward. This paves the way to explore in details the properties of Janus nanorods.

## Data availability

All data needed to evaluate the conclusions in the paper are present in the paper and/or the Supplementary Information. Additional data related to this paper may be requested from the corresponding authors.

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

## Acknowledgements

S.H. acknowledges financial support from the China Scholarship Council. G. Pembouong (IPCM, Paris) is acknowledged for assistance with SEC measurements. We thank Takasago company for the gift of β-pinene. Lazhar Benyahia is thanked for helpful discussions regarding the observation of the emulsions by confocal microscopy.

## Author contributions

O.C. and L.B. conceived the project. S.H. performed most experiments and analysed the data. D.Y. performed the stability study. C.L. and J.-M.G. performed the NMR and microscopy experiments, respectively, and analysed the data. F.N. performed the confocal microscopy experiments and analysed the data. S.P., J.R., and F.S. supervised the synthesis. J.J., E.N., O.C., and L.B. supervised the characterization. L.B. wrote the manuscript.

## Competing interests

The authors declare no competing interests.
