## [Peer Review File · Nature Communications]

REVIEWER COMMENTS

Reviewer #1 (Remarks to the Author):

In their article entitled "Straightforward preparation of supramolecular Janus nanorods by hydrogen bonding of end-functionalized polymers", the authors present the straightforward synthesis of Janus nanorods using functional RAFT chain transfer agents. The co-incorporation of two different "tagged" polymers into each construct is confirmed using DOSY NMR spectroscopy, SLS, SANS, and contrast-enhanced TEM imaging. The Janus nature of the anisotropic particles is then further exploited by preparing oil-in-water emulsions, where Janus samples act as more effective stabilizers than comparable homogenous nanorods or spherical micelles. Their approach represents a simple and elegant solution for the preparation of highly anisotropic structures, which hitherto has proven difficult at the nanometer length scale. However, despite the exciting implications of this work, the authors make several un-supported claims that act to undermine its quality. Further, no quantitative analysis or microscopy has been utilized to characterize the emulsions.

Issues/questions:

- The statement "it is therefore independent of the actual polymers used and works even for compatible polymers" is unfounded in the context of this work, as very few types of polymers are tested and no evidence is provided to suggest this method will be general. Indeed, it is likely the case that such a strategy would not be suitable for polymer chains that already possess charged or H-bonding groups.
- It is unclear from Fig. 1A how the red and blue chains are different. Further, the driving force for self-sorting should be further elaborated. It is clear that polymers have been attached to different "sides" of the trisurea "sticker"; however, it is not immediately obvious how this favors the type of alternating behavior depicted in Fig. 1A.
- Are the Janus nanorods shown in Figs. 1 and 2 at equilibrium, or were measurements taken directly after assembly? Do the nanorods evolve over time towards homo- or heterogeneous constructs? What happens if you mix two samples of pre-assembled homogenous nanorods together? Do they still form Janus structures?
- It is not clear in Fig. 2e that the nanorods have more contrast on one side compared with the other. It would be more convincing to include the grayscale analysis from the SI in the manuscript figure.
- No quantitative data or microscopy data for the oil-in-water emulsions has been provided. How do the droplet sizes differ between the various conditions? How stable are these droplet sizes over time?

Reviewer #2 (Remarks to the Author):

This paper describes straightforward preparation of supramolecular Janus nanorods by hydrogen bonding of end-functionalized polymers. The Janus nanorods are micrometer long and 10-nanometer wide. The Janus nanorods are synthesized by cooperative segregation of incompatible polymer arms and interactions between unsymmetrical and complementary hydrogen bonded stickers. As one applicative potential, it is demonstrated the Janus nanorods can efficiently stabilize oil-in-water emulsions. The

synthesis is interesting to derive Janus nanorods. Before considering acceptance, the following key points should be clearly addressed.

- 1) According to the conjecture, the Janus nanorods are formed mainly based on hydrogen bonding as illustrated in the Scheme. Please provide evidence they are in the form of rods rather than sheets. It is suggested to provide AFM image. It is imaging that there exists stronger steric hindrance. Therefore, the nanorods may possess helical structure. Please provide CD characterization.
- 2) What's the upper concentration limit for the Janus nanorods, therefore the yield of the product. What will happen in more concentrated systems. Are the product pure as mentioned, or other morphologies co-existent.
- 3) As illustrated for the potential application, the Janus nanorod can serve as a solid surfactant to stabilize emulsions. It is important to understand orientation of the Janus nanorod at the emulsion interface. What will happen for the orientation thus emulsion characteristic upon varying temperature or changing pH.

Thank you for your comments and suggestions. The changes made to the manuscript are detailed below.

Reply to Reviewer #1:

In their article entitled “Straightforward preparation of supramolecular Janus nanorods by hydrogen bonding of end-functionalized polymers”, the authors present the straightforward synthesis of Janus nanorods using functional RAFT chain transfer agents. The co-incorporation of two different “tagged” polymers into each construct is confirmed using DOSY NMR spectroscopy, SLS, SANS, and contrast-enhanced TEM imaging. The Janus nature of the anisotropic particles is then further exploited by preparing oil-in-water emulsions, where Janus samples act as more effective stabilizers than comparable homogenous nanorods or spherical micelles. Their approach represents a simple and elegant solution for the preparation of highly anisotropic structures, which hitherto has proven difficult at the nanometer length scale. However, despite the exciting implications of this work, the authors make several un-supported claims that act to undermine its quality. Further, no quantitative analysis or microscopy has been utilized to characterize the emulsions.

Issues/questions:

-The statement “it is therefore independent of the actual polymers used and works even for compatible polymers” is unfounded in the context of this work, as very few types of polymers are tested and no evidence is provided to suggest this method will be general. Indeed, it is likely the case that such a strategy would not be suitable for polymer chains that already possess charged or H-bonding groups.

Reply: We agree that our statement was too strong. It is obvious that the strategy will not work if the polymers are too bulky or compete with the stickers assembly. This part has been rephrased as: "Therefore, even compatible polymers can be used to form these Janus objects. In fact, any polymers should qualify, as long as they do not prevent co-assembly of the stickers."

-It is unclear from Fig. 1a how the red and blue chains are different. Further, the driving force for self-sorting should be further elaborated. It is clear that polymers have been attached to different “sides” of the trisurea “sticker”; however, it is not immediately obvious how this favors the type of alternating behavior depicted in Fig. 1a.

Reply: Fig 1a has been expanded to illustrate the instability of the competing structures. The difference between red and blue chains has been highlighted on Fig 1b.

-Are the Janus nanorods shown in Figs. 1 and 2 at equilibrium, or were measurements taken directly after assembly? Do the nanorods evolve over time towards homo- or heterogeneous constructs? What happens if you mix two samples of pre-assembled homogenous nanorods together? Do they still form Janus structures?

Reply: All samples were characterized after at least 12 hours of rest after assembly. The absence of long term evolution was checked and the data is now included in Supplementary Information (Fig. S27 and S28). Moreover, mixing two pre-assembled homogeneous nanorods does not allow co-assembly (data now shown in Fig. S26). These experiments indicate that the co-assemblies are most probably kinetically frozen.

-It is not clear in Fig. 2e that the nanorods have more contrast on one side compared with the other. It would be more convincing to include the grayscale analysis from the SI in the manuscript figure.

Reply: Grayscale analysis has now been included (Fig. 2f).

-No quantitative data or microscopy data for the oil-in-water emulsions has been provided. How do the droplet sizes differ between the various conditions? How stable are these droplet sizes over time?

Reply: The water/ethyl acetate emulsions have now been systematically monitored by confocal microscopy (Fig. S31). It shows that smaller droplets are obtained with the Janus nanorods compared to the homogeneous rods. Moreover, the coalescence of the droplets is strongly hindered between 1 and 9 days.

Reply to Reviewer #2:

This paper describes straightforward preparation of supramolecular Janus nanorods by hydrogen bonding of end-functionalized polymers. The Janus nanorods are micrometer long and 10-nanometer wide. The Janus nanorods are synthesized by cooperative segregation of incompatible polymer arms and interactions between unsymmetrical and complementary hydrogen bonded stickers. As one applicative potential, it is demonstrated the Janus nanorods can efficiently stabilize oil-in-water emulsions. The synthesis is interesting to derive Janus nanorods. Before considering acceptance, the following key points should be clearly addressed.

1) According to the conjecture, the Janus nanorods are formed mainly based on hydrogen bonding as illustrated in the Scheme. Please provide evidence they are in the form of rods rather than sheets. It is suggested to provide AFM image. It is imaging that there exists stronger steric hindrance. Therefore, the nanorods may possess helical structure. Please provide CD characterization.

Reply:

Sheets: Our attempts to analyse the particles by AFM on silica substrates have been unsuccessful. However, no sheets (2D-objects) were detected either by cryo-TEM or by conventional TEM (more than 100 pictures where only rods (1D-objects) were seen). Moreover, SANS and light scattering should show a q^{-2} behaviour at low q if sheets were present, even in low amount. This is not the case: the q^{-1} dependence of the scattered intensity recorded by SANS is a clear evidence that rods are formed rather than sheets. Actually, scattering provides a global picture of a sample, in contrast to microscopy which provides local snapshots. Therefore, based on the scattering data, we can reasonably rule out the presence of significant amounts of 2D, sheet-like objects. Of course, it does not mean that the cross-section of the rods is circular. In fact, the fit of the high q SANS data indicates that the cross-section is elliptical (Fig. 2b), with an aspect ratio of $12/7.6=1.6$, i.e. ribbon-like.

Helicity: The solution of (4/8-PDMAc + 8/4-PNAM) Janus nanorods has proved to be CD-silent, but this absence of signal is not conclusive since no chiral bias is present in the system. The contrast-enhanced microscopy data (Fig. S24) shows very few objects with a darkness modulation along the rod axis. These modulations are consistent with a very low amount of twisting of the rods, which can be due either to a long-range flexibility of the rods or the presence of helicity. However, even if this is the sign of a helical nature of the objects, their pitch is apparently very large (at least on the order of 100 nm). The presence of some long-distance twisting has been highlighted in the manuscript.

2) What's the upper concentration limit for the Janus nanorods, therefore the yield of the product. What will happen in more concentrated systems. Are the product pure as mentioned, or other morphologies co-existent.

Reply: In the manuscript, two preparation methods are described.

1) The direct preparation consists in injecting water in a 100 g/L solution in DMSO, so that the final polymer concentration is 1 g/L. This is more or less the highest concentration feasible by this procedure. This limit has now been mentioned in the SI. It is difficult to assess the purity. Indeed,

Sorbonne Université

4, place Jussieu, 75252 Paris cedex 05

barre 43-53, 5^{ème} étage, 510, case courrier 185

Tel: 33 (0)1 44 27 61 79 / Fax: 33 (0)1 44 27 55 04

laurent.bouteiller@upmc.fr

<http://www.ipcm.fr/-polymeres->

all the characterizations performed (NMR, LS, SANS, TEM) prove that the main product of the assembly is the Janus nanorod. Of course, even if they were not detected, we cannot rule out the presence of a minor content of homo-assemblies.

2) The second preparation consists in isolating the previously prepared particles as a solid powder. The yield of this step is 80-90 %, due to material loss when transferring the solid powder (the information has now been added in SI). From this powder, a new dispersion can be prepared in water at any desired concentration.

3) *As illustrated for the potential application, the Janus nanorod can serve as a solid surfactant to stabilize emulsions. It is important to understand orientation of the Janus nanorod at the emulsion interface. What will happen for the orientation thus emulsion characteristic upon varying temperature or changing pH.*

Reply: We assume that the most hydrophilic part of the Janus nanorod is oriented towards the water phase, but this is difficult to prove. However, we have tested the stability of the particles to temperature and pH. In water/DMSO (99/1) dispersions, the particles are remarkably stable, up to 80°C (Fig. S29) and pH=10 (Fig. S30).

We have also tested the stability of the water/ethyl acetate emulsions to pH, but the results are inconclusive due to hydrolysis of ethyl acetate.

Finally, we tested the stability of the water/ethyl acetate emulsions at 50°C and observed coalescence of the ethyl acetate droplets within 1 hour. This phenomenon is reversible: re-emulsification is possible upon cooling the system. Although this result is quite interesting, it may originate from several possibilities: variation of the affinity of one or both blocks for the interface, increased thermal energy altering the adhesion of the particles at the interface or partial dissociation of the particles (this latter possibility seems to be ruled out by the temperature stability experiment in water, but might occur in the presence of ethyl acetate). Understanding rigorously this result deserves further investigation that lies out of the scope of the present paper. We nevertheless thank the reviewer for this idea which brings new research opportunities for this system.

REVIEWERS' COMMENTS:

Reviewer #1 (Remarks to the Author):

My comments have now been fully addressed and I am happy for the paper to be accepted.

Reviewer #2 (Remarks to the Author):

I think the revised manuscript can be published now.